# Attractive and Repulsive Perceptual Biases Naturally Emerge in Generative Adversarial Inference

## Abstract

Perceptual estimates exhibit a reversal in bias depending on uncertainty: they shift toward prior expectations under high stimulus noise, but away from them when sensory noise dominates. The normative framework of a Bayesian observer model can account for this phenomenon, yet most formulations treat it as given rather than explaining its emergence through learning. We introduce a Generative Adversarial Inference (GAI) network that acquires latent representations and inference strategies directly from sensory inputs, without hand-crafted likelihoods or priors. Trained using adversarial learning with reconstruction on Gabor stimuli under varying uncertainty, the network learns to recover underlying stimuli from noisy inputs, and spontaneously reproduces the bias reversal observed in human perception. This emergent behavior arises from network responses that reveal signatures of efficient coding and Bayesian inference. Our findings provide an end-to-end account of perceptual bias that unifies normative theory and deep learning.

## 1 Introduction

Human perceptual estimates are systematically biased. Sometimes they are pulled toward statistically frequent values, producing attractive biases, while in other cases they are pushed away, producing repulsive biases. For example, orientation estimation shows attraction toward and repulsion away from the cardinal axes, horizontal and vertical, dominant components in natural scenes (Bouma & Andriessen, 1968; Girshick et al., 2011; De Gardelle et al., 2010; Tomassini et al., 2010; Sun et al., 2025). The coexistence of attraction and repulsion may at first seem contradictory. Yet these patterns are not arbitrary mistakes or simple heuristics, but lawful signatures of how the brain encodes sensory inputs and infers their causes.

These biases can be seen as the best guesses the brain can make given biological limitations. Sensory signals are noisy and incomplete, and the brain combines them with prior expectations to infer their most likely causes (Ernst & Banks, 2002; Knill & Pouget, 2004). In this framework, uncertainty dynamically modulates the balance between sensory evidence and prior expectations (Kwon et al., 2015; Wei & Stocker, 2015). Crucially, perceptual inference necessarily starts with signals encoded by the sensory system. Under the efficient coding hypothesis (Barlow, 1961; Attneave, 1954), these representations are inevitably shaped by environmental statistics: to use limited neural resources efficiently, more are devoted to frequent stimuli, and the sensory space becomes warped. In such a distorted space, internal noise produces characteristic biases in estimation. When Bayesian inference operates on these representations, both attraction toward frequent values and repulsion away from them emerge naturally, as demonstrated in recent studies (Wei & Stocker, 2015; Hahn & Wei, 2024).

Bayesian observer models have been highly influential, offering a clear framework for perception under uncertainty. Yet they remain symbolic in nature: built on handcrafted priors and likelihoods, they leave key questions unresolved. They capture perceptual biases, but not their emergence through learning. They also offer little account of how priors and likelihoods are acquired and represented in neural systems. Perception, however, is not a sequence of isolated estimates but a continuous reconstruction of reality (Clark, 2013). This calls for models that learn representation and inference directly from data. Connectionist models can exhibit warped representations under specific training conditions (Benjamin et al., 2022), but achieving neural plausibility has been difficult, as these models

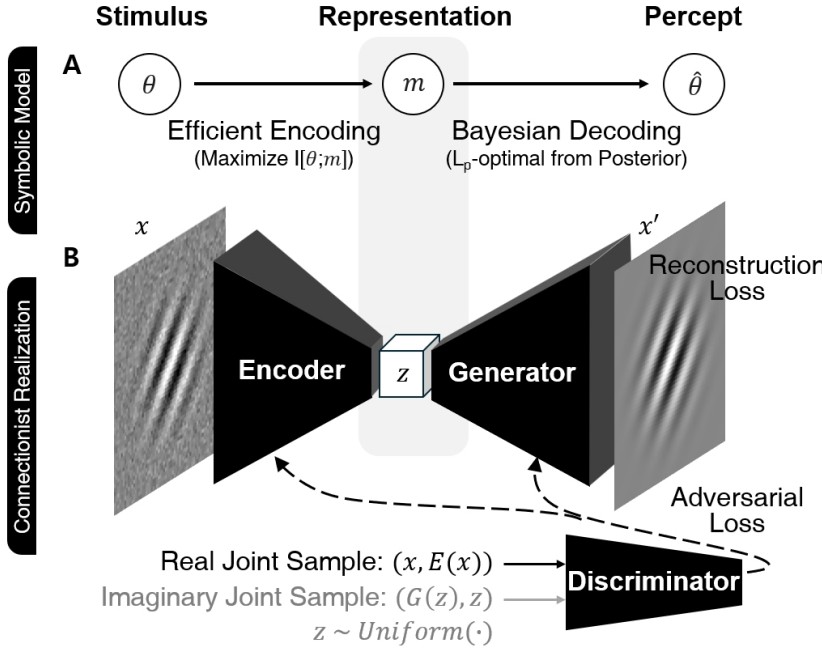

Figure 1: Symbolic vs. generative adversarial inference. (A) Symbolic model: $\theta$ is efficiently encoded into a noisy measurement $m$, and inference recovers $\hat{\theta}$ from the posterior. (B) Generative Adversarial Inference (GAI): $E$ maps $x$ to $z$, and $G$ reconstructs $x'$. Adversarial training aligns $(x, z)$ pairs, while a reconstruction loss enforces input–output consistency. The internal representation reflects stimulus statistics, yielding efficient coding and implicit Bayesian inference.

operate mainly as feedforward machines trained to match correct answers, rather than systems that learn inference.

A more compelling proposal comes from Gershman's "generative adversarial brain" perspective (Gershman, 2019), which frames perception as the interaction between an encoder that maps sensory inputs into internal representations and a generator that reconstructs those inputs. In this view, bottom-up signals from the encoder and top-down predictions from the generator are jointly constrained to maintain consistency. This bidirectional negotiation mirrors predictive coding accounts of perception (Rao & Ballard, 1999; Friston, 2010), offering a biologically plausible rationale for adversarial architectures. Gershman's proposal has so far remained at the level of perspective. While adversarial architectures such as Bidirectional Generative Adversarial Networks (BiGANs; Donahue et al. 2017) provide structural realizations of encoder–generator consistency, they have not been applied to explain human perceptual behavior or tested against psychophysical data. More importantly, they lack any account of uncertainty, a factor central to human perception and critical for explaining the reversal from attraction to repulsion.

To address this gap, we introduce Generative Adversarial Inference (GAI), a learned system in which bias reversals emerge naturally. GAI instantiates the adversarial framework with denoising and constrained latent resources. Through adversarial training, representation and inference are shaped together: the system recovers stimuli from noisy inputs while reconstructions are tested against the input distribution. In this way, GAI learns efficient representations and an implicit inference strategy that mirror human perceptual biases. A single learned system thus accounts for both efficient coding and adaptive inference, bridging Bayesian and connectionist perspectives.

**Contributions.**

- Introduce GAI, a learned system that reproduces both attractive and repulsive perceptual biases without hand-crafted priors and likelihoods.
- Show that these biases emerge from efficient coding combined with Bayesian inference, and that the model produces stimulus-level reconstructions beyond point estimates.

## 2 BACKGROUND & RELATED WORK

### 2.1 ORIENTATION BIASES: EVIDENCE AND BAYESIAN ACCOUNTS

Orientation perception consistently shows systematic biases. When external noise dominates (e.g., when multiple Gabor elements are jittered within an ensemble), perceived orientations are drawn toward the cardinal axes. This attractive bias is often several degrees in magnitude, ranging from about 3° to more than 10° across studies (Bouma & Andriessen, 1968; Girshick et al., 2011). Such attraction is consistent with the role of cardinals as strong perceptual priors, supported by analyses of natural images showing a predominance of horizontal and vertical orientations (Girshick et al., 2011). By contrast, when internal noise dominates, such as under low contrast stimuli, orientation estimates shift away from the cardinals. Repulsive errors are typically between 2° and 7° across studies (De Gardelle et al., 2010; Tomassini et al., 2010; Sun et al., 2025). Both attractive and repulsive biases have been observed across diverse paradigms, suggesting that they reflect fundamental properties of orientation perception rather than task-specific artifacts.

These robust patterns of attraction and repulsion have motivated normative models of perception. Bayesian observer models provide a natural account of how uncertainty shapes perception (Knill & Pouget, 2004; Ernst & Banks, 2002). By combining noisy sensory evidence with prior expectations, they explain why estimates tend to regress toward frequently occurring values. Formally, perception can be cast as Bayesian inference:

$$p(\theta \mid m) \propto p(m \mid \theta)\, p(\theta),$$

where $m$ denotes the noisy sensory measurement and $p(\theta)$ the prior distribution. For illustration, consider the simple case of a Gaussian likelihood, $p(m \mid \theta) = \mathcal{N}(m; \theta, \sigma_m^2)$, and a Gaussian prior, $p(\theta) = \mathcal{N}(\theta; \mu_p, \sigma_p^2)$. The posterior is also Gaussian with optimal estimate

$$\hat{\theta} \;=\; \frac{\lambda_m m + \lambda_p \mu_p}{\lambda_m + \lambda_p},$$

where $\lambda_m = 1/\sigma_m^2$ and $\lambda_p = 1/\sigma_p^2$ denote the respective precisions, showing that $\hat{\theta}$ dynamically shifts with the relative reliabilities of likelihood and prior.

Wei & Stocker (2015) extended this framework with efficient coding, assuming limited neural resources that are allocated in proportion to stimulus frequency. The Fisher information of the sensory representation is defined as

$$J(\theta) \;=\; \mathbb{E}_{m \sim p(m|\theta)}\!\left[\left(\frac{\partial}{\partial \theta} \log p(m \mid \theta)\right)^2\right].$$

Under the efficient coding hypothesis, it is linked to the stimulus prior and the discrimination threshold:

$$\sqrt{J(\theta)} \propto p(\theta), \qquad D(\theta) \propto \frac{1}{\sqrt{J(\theta)}},$$

where $D(\theta)$ is the discrimination threshold, and this relationship is formally derived in Wei & Stocker (2016). As a result, common orientations are encoded with higher fidelity and discriminated more precisely. When Bayesian inference operates on such warped representations, likelihood asymmetries emerge: external noise produces attraction toward the prior, while internal noise produces repulsion away from it. This framework provides a principled account of bias reversals, but it remains symbolic. Whether such biases can emerge naturally through learning is unknown.

### 2.2 EMERGENT PERCEPTUAL STRUCTURE IN NEURAL NETWORKS

Normative models show how bias reversals could in principle arise, but an open question is whether such structure can also emerge in learned systems. Deep networks provide a test case: when optimized on naturalistic tasks, they often develop internal codes with perceptual signatures.

For example, Rajalingham et al. (2022) trained recurrent networks on a physical inference task that required predicting occluded trajectories, and found that hidden states spontaneously encoded velocity information. Rideaux & Welchman (2020) introduced a convolutional model (MotionNet) trained

to classify motion direction from natural video input, which developed anisotropic direction tuning resembling human motion biases. Nasr et al. (2019) showed that a CNN trained for number classification produced units tuned to numerosity with Weber–Fechner scaling, echoing psychophysical laws. Farzmahdi et al. (2024) reported that CNNs trained on object recognition develop mirror-symmetric viewpoint tuning. These examples indicate that perceptual structure can emerge as a byproduct of task-driven learning, without explicit design.

Gradient-based learning has also been linked to efficient-like representations. Henderson & Serences (2021) demonstrated that orientation discriminability in VGG-16 reflected the statistics of rotated ImageNet images. Benjamin et al. (2022) further showed that networks trained to reconstruct images from compressed codes initially allocate more representational resources to frequent features, yielding Fisher-information profiles aligned with input statistics. However, such asymmetries typically appear only under specific training regimes and tend to diminish as networks fully converge, making the effect transient.

Related signatures have been observed in isolation—bias attraction (Rideaux & Welchman, 2020) and Fisher-information warping (Henderson & Serences, 2021; Benjamin et al., 2022; Mao et al., 2025). What remains unclear is whether a single learned system can sustain efficient representations while also producing the uncertainty-dependent biases observed in human perception.

## 2.3 LIMITATIONS AND OUR APPROACH

The "generative adversarial brain" perspective (Gershman, 2019) casts perception as a negotiation between bottom-up encoding and top-down generation, reminiscent of predictive coding theories (Rao & Ballard, 1999; Friston, 2010). BiGANs (Donahue et al., 2017) instantiate this idea: an encoder maps inputs to a latent representation, a generator reconstructs from it, and a discriminator enforces joint alignment by distinguishing real pairs $(x, E(x))$ from synthetic pairs $(G(z), z)$. This training enforces consistency between sensory inputs and internal representations. Whereas variational or adversarial autoencoders optimize marginal objectives, BiGANs directly align the joint distribution $(x, z)$, yielding a tighter coupling between data and representation.

However, these models do not account for an essential factor of perceptual processing—uncertainty in inputs and representations—and thus no existing framework unifies efficient representation with uncertainty-sensitive inference in a single learned system.

## 3 GENERATIVE ADVERSARIAL INFERENCE (GAI)

We introduce Generative Adversarial Inference (GAI), a learnable encoder–generator–discriminator architecture trained end-to-end with a joint adversarial–reconstruction objective. GAI extends the adversarial framework with denoising, allowing the encoder to form internal representations that adapt to stimulus statistics while the generator reconstructs inputs from noisy observations. In this way, the model acquires an implicit inference strategy without requiring explicit likelihoods or priors, and it reproduces the reversal of bias direction with changing noise conditions, behaving as if performing Bayesian inference under uncertainty.

### 3.1 JOINT OBJECTIVE

We train the encoder $E$, generator $G$, and discriminator $D$ with a joint adversarial–reconstruction loss:

$$\min_{E,G} \max_{D} \; L_{\text{adv}}(E, G, D) + w_{\text{recon}} L_{\text{recon}}(E, G). \tag{1}$$

The adversarial term is

$$
\begin{aligned}
L_{\text{adv}} = \; & \mathbb{E}_{x \sim p_{\text{data}}}\big[D(x, E(x))\big] - \mathbb{E}_{z \sim p(z)}\big[D(G(z), z)\big] \\
& + \lambda \, \mathbb{E}_{\hat{x}, \hat{z}}\Big[\big(\|\nabla D(\hat{x}, \hat{z})\|_2 - 1\big)^2\Big],
\end{aligned}
\tag{2}
$$

where $D$ is optimized with the WGAN-GP objective to stabilize training and provide a continuous loss. Here, $\lambda$ controls the strength of the gradient penalty ($\lambda = 20$).

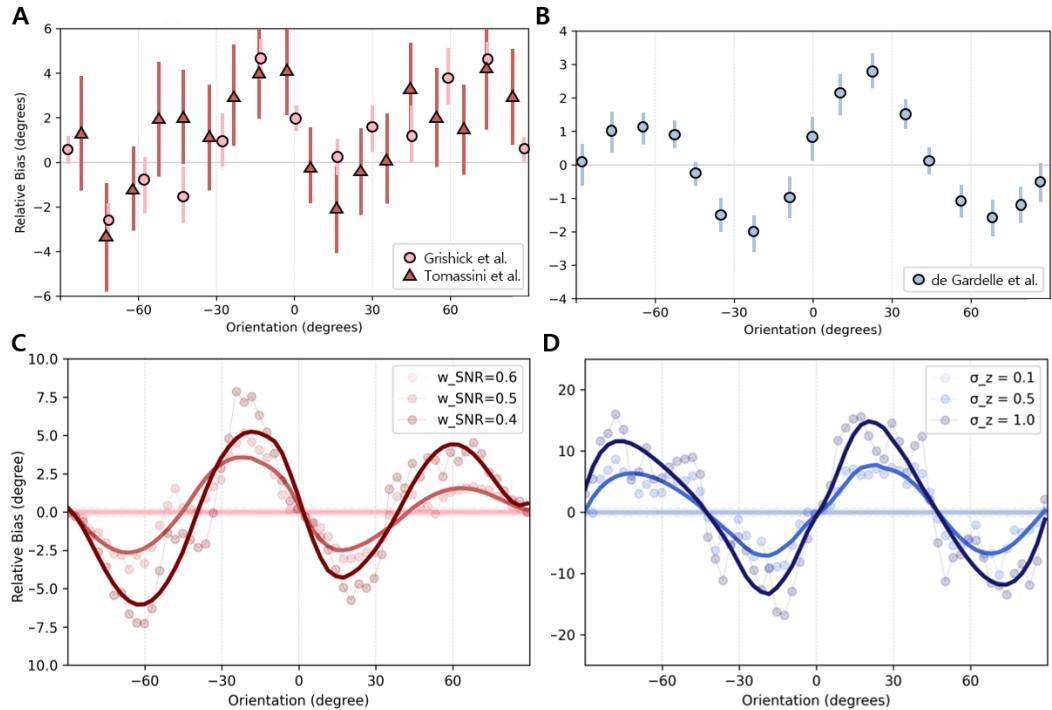

Figure 2: Relative bias under external vs. internal noise. (A) Human data replotted from Wei & Stocker (2015), based on orientation estimates reported in Girshick et al. (2011) and Tomassini et al. (2010): increasing *stimulus* noise produces stronger attraction toward the prior mean. (B) Human data replotted from Wei & Stocker (2015), based on De Gardelle et al. (2010): increasing *sensory* (internal) noise produces repulsion away from the prior mean. (C) GAI model under external noise: for each orientation, multiple noisy Gabors were passed through the encoder–decoder, and orientations decoded from reconstructions were averaged (scatter). (D) GAI under internal noise: clean inputs were encoded once, latent codes perturbed with Gaussian noise, and decoded orientations averaged (scatter). Bold curves in C–D show smoothed trends obtained with a Savitzky–Golay filter.

The reconstruction term is

$$L_{\text{recon}} = \mathbb{E}_{x \sim p_{\text{data}}} \|x - G(E(x))\|_1, \tag{3}$$

with $w_{\text{recon}}$ weighting the importance of input–output consistency ($w_{\text{recon}} = 8$).

## 3.2 DATA GENERATION AND EVALUATION PROTOCOL

We generated grayscale Gabor patches with orientations drawn from a bimodal prior favoring the cardinal axes (0° and 90°). Each patch was corrupted by additive Gaussian noise, with the signal-to-noise ratio varied continuously by a weighted mixture of signal and noise. Noisy inputs contained a random mixture of 0° and 90° phase components. Full architecture, training hyperparameters, and data generation details are provided in Appendix A. The reported effects were robust across multiple random seeds, yielding qualitatively consistent results.

## 4 RESULTS

We next show that GAI reproduces the bias reversal observed in human perception and examine ablation models to isolate the role of each component.

### 4.1 BIAS REVERSAL WITH STIMULUS UNCERTAINTY

We first asked whether GAI reproduces the hallmark reversal between attractive and repulsive biases predicted by efficient Bayesian observer models. Figures 2A–B replot human data as summarized in

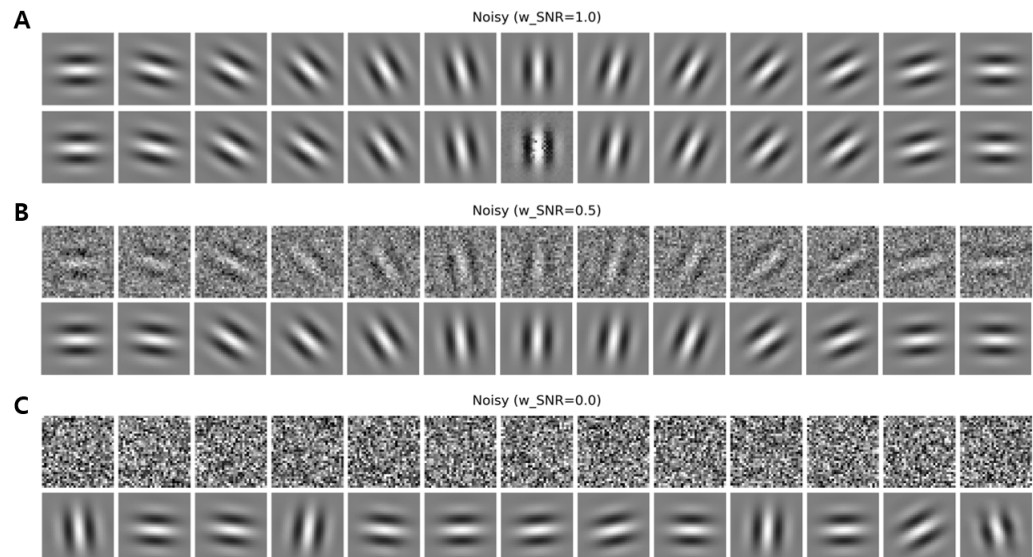

Figure 3: Attractive bias emerges with increasing stimulus noise. (A) Reconstructions when no external noise is added: outputs closely match inputs across orientations. (B) With moderate stimulus noise, reconstructions begin to concentrate toward the cardinal axes, reflecting prior attraction. (C) With maximal stimulus noise, reconstructions collapse almost entirely to the cardinals, showing strong attractive bias. Panels illustrate representative input Gabor patches (first rows) and corresponding reconstructions by the GAI model (second rows).

Wei & Stocker (2015): when stimulus noise is increased (external corruption of the input), estimates are pulled toward the prior mean (Girshick et al., 2011; Tomassini et al., 2010); when variability instead reflects sensory noise, estimates are repelled away from the prior (De Gardelle et al., 2010). These complementary patterns establish the empirical benchmark that any model must account for.

Our model recapitulates both patterns (Figure 2C–D). Under high external noise, reconstructions converge toward the cardinal axes, consistent with prior attraction. In contrast, when Gaussian noise is injected into the latent space, estimates shift away from the cardinals, revealing systematic repulsion. These complementary effects show that a single learned system can account for both sides of the bias reversal.

The effect of external noise is further illustrated in Figure 3. With clean inputs, reconstructions align closely with the true orientation. As stimulus noise increases, outputs increasingly concentrate toward the cardinal orientations, and under maximal noise they collapse almost entirely to the prior peaks. This central-tendency bias emerges spontaneously, reflecting the internalized statistics of the observed data.

Together, these results indicate that GAI reproduces the qualitative bias reversal and shows an uncertainty dependence consistent with Bayesian observer accounts.

### 4.2 PRIOR AND EFFICIENT CODING IN INTERNAL REPRESENTATION

To understand the source of these biases, we next examined how the model's internal representation reflects the statistics of the training environment. Figure 4A shows the empirical orientation histogram of training stimuli, highlighting the predominance of the cardinal axes. When the model is presented with maximally noisy inputs, its reconstructions collapse to the same peaks (Figure 4B), revealing that GAI has internalized this bimodal prior through learning rather than explicit specification.

We then quantified representational precision by computing the square root of Fisher information from encoder Jacobians (Figure 4C). The profile exhibits clear peaks at the cardinals, closely matching the stimulus prior. This efficient-coding signature indicates that the encoder allocates

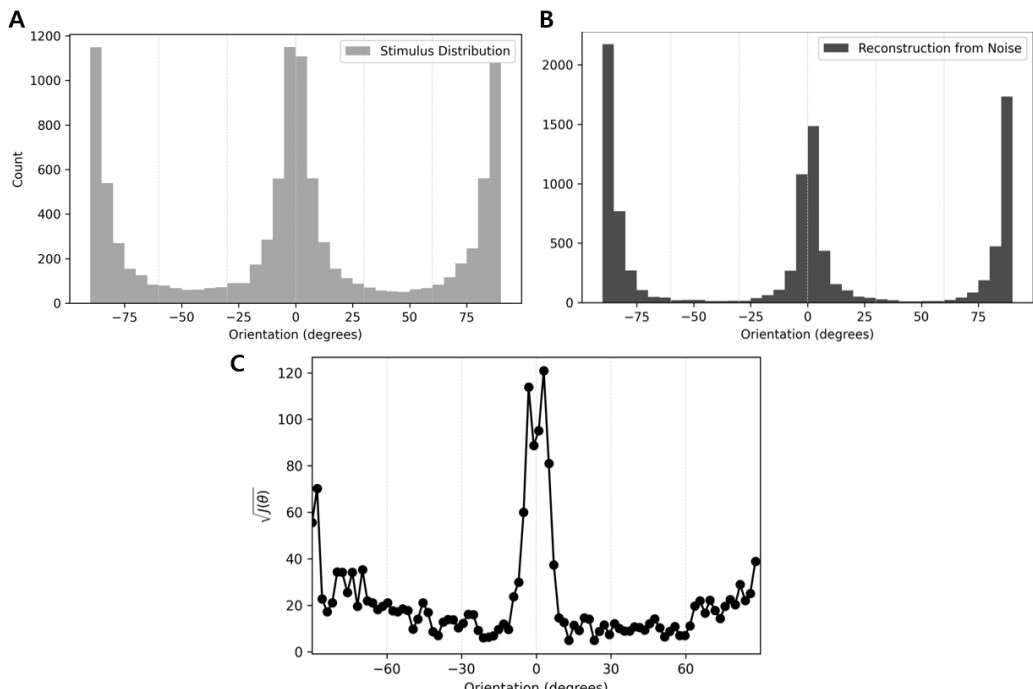

Figure 4: Internal representations reflect orientation priors and efficient coding. (A) Orientation histogram of training stimuli, showing cardinal dominance. (B) Orientation histogram of reconstructions from maximally noisy inputs, revealing the model's learned prior. (C) Square root Fisher information across orientations, computed from encoder Jacobians, showing higher precision at cardinals.

greater representational resources to more frequent orientations, in line with normative theory. The orientation space is thus warped by environmental statistics, which explains why internal noise injected into the latent space yields systematic repulsion: estimates are pushed away from regions of high representational precision.

These analyses show that GAI acquires both a learned prior and an efficient code through data-driven training. The interaction of these two elements—statistical priors and resource allocation—forms the basis for the observed reversal between attractive and repulsive biases. We next test which components of the training objective are necessary for these properties using ablation models.

## 4.3 ABLATION

### 4.3.1 RECONSTRUCTION-ONLY

To test whether adversarial training is necessary for shaping the orientation manifold, we trained a reconstruction-only variant of the model. This baseline recovered input–output consistency but lacked the adversarial alignment between latent codes and the data distribution.

As shown in Appendix Fig A1, the baseline produces a weak attractive trend under stimulus noise, but the effect is much less systematic than in the full model. More importantly, repulsive biases fail to emerge. Instead of smooth shifts away from the cardinals, the bias profile is dominated by irregular bumps, indicating that internal variability is not transformed into lawful repulsion.

Inspection of latent samples further highlights the deficit. Whereas the full model generates diverse but structured reconstructions along the orientation manifold, the reconstruction-only baseline often produces hallucinated patterns that deviate from the trained stimuli. This instability reflects the absence of a well-formed latent geometry.

These results suggest that adversarial training plays a critical role in stabilizing the internal representation. Without it, the encoder cannot sustain an efficient code that supports systematic repulsion under latent noise.

### 4.3.2 ADVERSARIAL-ONLY

We also tested an adversarial-only variant, trained without the reconstruction term. This model preserved adversarial alignment between latent codes and the data distribution but lacked an explicit input–output consistency constraint.

As shown in Appendix Fig. A2, residual attractive trends can be observed under stimulus noise, but repulsive biases fail to emerge systematically. Under maximally noisy inputs, reconstructed orientations and Gabor images both degenerate to horizontal patterns, indicating a strong mode collapse. This suggests that adversarial loss alone may be vulnerable to reduced diversity in reconstructions and to an unstable orientation manifold.

Taken together, the two ablations indicate complementary roles of each component: reconstruction loss supports stable input–output mapping, while adversarial training helps shape the latent geometry. Their combination appears necessary to reproduce both attractive and repulsive biases in a manner consistent with human perception.

## 5 DISCUSSION

Our findings show that both attractive and repulsive biases arise within a single learned system, as a direct consequence of efficient coding and Bayesian inference. In contrast to previous studies that only reported efficient-like representations, GAI establishes a direct link between representational asymmetries and behavioral biases. This provides a concrete computational instantiation of the "generative-adversarial brain" hypothesis, unifying normative theory with learnable neural architectures.

### 5.1 ROLE OF ADVERSARIAL TRAINING IN REPRESENTATION

The reconstruction-only model can reproduce inputs but does not learn to organize the latent space in a way that reflects the full distribution. As a result, internal variability is not expressed as systematic repulsion but as irregular deviations, showing that the internal space is only partially exploited. With adversarial training, latent codes are aligned with the input distribution, the orientation manifold becomes smooth, and perturbations in latent space give rise to lawful repulsive shifts.

Adversarial loss is therefore essential not because it improves reconstruction per se, but because it allows the model to fully utilize its internal space. Only then can both attraction and repulsion emerge within a single system.

### 5.2 DEFINITION OF EXTERNAL NOISE

Our definition of external noise follows the framework of (Lu & Dosher, 1998), who distinguished between internal and external sources of variability. Consistent with this definition, we implemented external noise as additive pixel noise on single Gabor patches. This choice differs in implementation from studies on cardinal biases that used ensemble stimuli with orientation jitter (Tomassini et al., 2010; Girshick et al., 2011), although both approaches target external sources of variability. We adopted the simpler form here because it fits our modeling framework and directly implements the original definition of external noise. Extending the model to ensemble-based stimuli will require additional mechanisms for pooling across elements, which we see as an important direction for future work.

### 5.3 LIMITATIONS AND FUTURE WORK

Our study has several limitations. The model was trained only on Gabor patches varying along a single orientation dimension, and the orientation manifold emerged without explicit circular constraints. Extending the framework to motion, color, or natural images is needed to test generality. We

implemented external noise as pixel-level corruption on single Gabors, whereas psychophysical studies often use ensemble stimuli with orientation jitter, as described earlier. Extending the model to such paradigms remains to be done. Finally, our evaluation relied mainly on qualitative comparison to a Bayesian observer model. Direct quantitative benchmarks against normative predictions are still required.

## 6 BROADER IMPACT

This work advances our understanding of how perceptual biases emerge from efficient coding and inference, bridging computational neuroscience and deep learning. By providing an end-to-end account of human-like biases, it offers insights for building AI systems that better align with human perception and for developing models to study sensory disorders. Potential applications include more interpretable machine learning algorithms and tools for neuroscience research.

## 7 CONCLUSION

We introduced Generative Adversarial Inference, a framework that learns perceptual representations and inference strategies directly from sensory inputs. Without explicit priors or likelihoods, the model reproduces the reversal of bias direction with changing uncertainty, consistent with efficient coding and Bayesian inference. These findings provide a unified account of attractive and repulsive perceptual biases, linking normative theory with learnable neural architectures. Beyond modeling human perception, our framework may also help analyze artificial systems, where comparable biases can emerge in generative models.

We will release all code and data upon publication.

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

## A APPENDIX: ARCHITECTURE AND TRAINING DETAILS

### A.1 NETWORK ARCHITECTURE

The encoder $E$ consists of four convolutional layers with $4 \times 4$ kernels and stride 2. The number of output channels is $\{64, 128, 256, 256\}$, each followed by batch normalization and ReLU activation. A final $1 \times 1$ convolution maps the output to a 5-dimensional latent vector $z \in \mathbb{R}^5$.

The generator $G$ mirrors the encoder with transposed convolutions. Latent vectors are first projected to a $4 \times 4$ spatial map. Four ConvTranspose2d layers with output channels $\{256, 128, 64, 1\}$ and

stride 2 progressively upsample this map to reconstruct $32 \times 32$ images. ReLU activations are used in all hidden layers and tanh at the output.

The discriminator $D$ has two branches. The image branch processes either real images $x$ or generated samples $G(z)$ with three convolutional layers (kernel size $4 \times 4$, stride 2, channels $\{64, 128, 256\}$) and LeakyReLU activations. The latent branch maps $z$ (or $E(x)$) through two fully connected layers with 128 hidden units and LeakyReLU activations. Outputs from the two branches are concatenated and passed through two fully connected layers (256 hidden units, LeakyReLU) for binary classification.

The overall processing pipeline can be summarized as:
$$x_{\text{noisy}} \xrightarrow{E} z \xrightarrow{G} \hat{x}_{\text{true}} \xrightarrow{D_\theta} \hat{\theta}^x.$$

The adversarial objective compares real and synthetic pairs:
$$(x_{\text{true}}, E(x_{\text{noisy}})) \quad \text{vs.} \quad (G(z), z), \quad z \sim p(z).$$

## A.2   TRAINING DETAILS

We trained all models with Adam (learning rate $1 \times 10^{-4}$, $\beta_1 = 0.5$, $\beta_2 = 0.9$) for 20,000 iterations, batch size 64. The discriminator was updated three times per encoder–generator step. Latent codes were sampled from a uniform distribution with unit variance. We used 20,000 synthetic Gabor patches for training; details of data generation and prior specification are described in Sec. 3.2. All numerical hyperparameters (e.g., learning rate, batch size, update ratio) were empirically chosen for stable training rather than exhaustively tuned.

## A.3   GABOR PATCH GENERATION AND PRIOR DISTRIBUTION

Each stimulus was defined as a two-dimensional Gabor function:
$$g(x, y) = \exp\left(-\frac{x'^2 + y'^2}{2\sigma^2}\right) \cos\left(2\pi f x' + \phi\right),$$
where
$$x' = x \cos\theta + y \sin\theta, \quad y' = -x \sin\theta + y \cos\theta.$$
Parameters were fixed as spatial frequency $f = 0.1$ cycles/pixel, Gaussian envelope $\sigma = 6$ pixels, and phase $\phi \in \{0°, 90°\}$ chosen randomly for each sample. Stimuli were rendered at $32 \times 32$ resolution.

*Prior distribution.* Orientations $\theta$ were drawn from a bimodal distribution favoring the cardinal axes. Specifically, we sampled from two wrapped Cauchy components centered at $0°$ and $90°$ with concentration parameter $\kappa = 0.1$, then remapped values into $[-90°, 90°]$. This procedure yields a prior distribution with two peaks at the cardinals, matching the statistics shown in Figure 4A.

*Noise corruption.* Additive Gaussian noise was applied at varying signal-to-noise ratios by linearly mixing clean Gabor patches with Gaussian noise fields:
$$x_{\text{noisy}} = w_{\text{SNR}} x_{\text{true}} + (1 - w_{\text{SNR}}) \eta,$$
where $\eta \sim \mathcal{N}(0, 1)$ and $w_{\text{SNR}} \in [0, 1]$ controlled the effective SNR.

## A.4   DECODER FOR ORIENTATION ESTIMATION

To decode orientation from reconstructed images, we implemented a bank-of-Gabor decoder. Specifically, for each candidate orientation $\theta \in [-90°, 90°]$, we generated a template Gabor patch $g_\theta$ of the same size and frequency as the training stimuli. Given a reconstructed image $\hat{x}_{\text{true}} = G(z)$, we computed the mean squared error (MSE) between $\hat{x}_{\text{true}}$ and each template:
$$\text{MSE}(\theta) = \frac{1}{N} \sum_{i=1}^{N} \left(\hat{x}_{\text{true}}(i) - g_\theta(i)\right)^2.$$

The decoder $D_\theta$ selects the orientation $\hat{\theta}^x$ that minimizes this error:
$$\hat{\theta}^x = \arg\min_\theta \ \text{MSE}(\theta).$$

In other words, $D_\theta$ serves as a template-matching decoder that estimates the most likely orientation by comparing the reconstructed image against a bank of oriented Gabors.

### A.5 Bias estimation procedure for Figures 2 and A1–A2

To quantify orientation biases under external and internal noise, we used the following procedure. For the *external noise* condition (Figure 2C), we generated 300 noisy Gabor patches per orientation by mixing clean stimuli with Gaussian noise at the specified $w_{SNR}$. Each noisy input was passed through the encoder–decoder pipeline, and reconstructed images were decoded with the template-matching procedure described in Appendix A.4. The mean decoded orientation across 300 samples provided one scatter point per input orientation.

For the *internal noise* condition (Figure 2D), clean inputs were first encoded once to obtain latent representations. Each latent vector was then perturbed with independent multivariate Gaussian noise to generate 300 samples, which were decoded to images and processed with the same orientation decoder. The mean decoded orientation across samples was taken as the scatter point.

The same procedure was applied in the ablation analyses. Specifically, Figures A1A and A2A test attractive bias under external noise, while Figures A1B and A2B test repulsive bias under internal noise.

In all cases, scatter points were further smoothed to highlight systematic trends. For Figures 2C–D we applied a Savitzky–Golay filter (window length 31, polynomial order 5). For the ablation results (Figures A1–A2), a shorter window length of 11 was used to better capture local trends.

## B    Appendix: Ablation Results

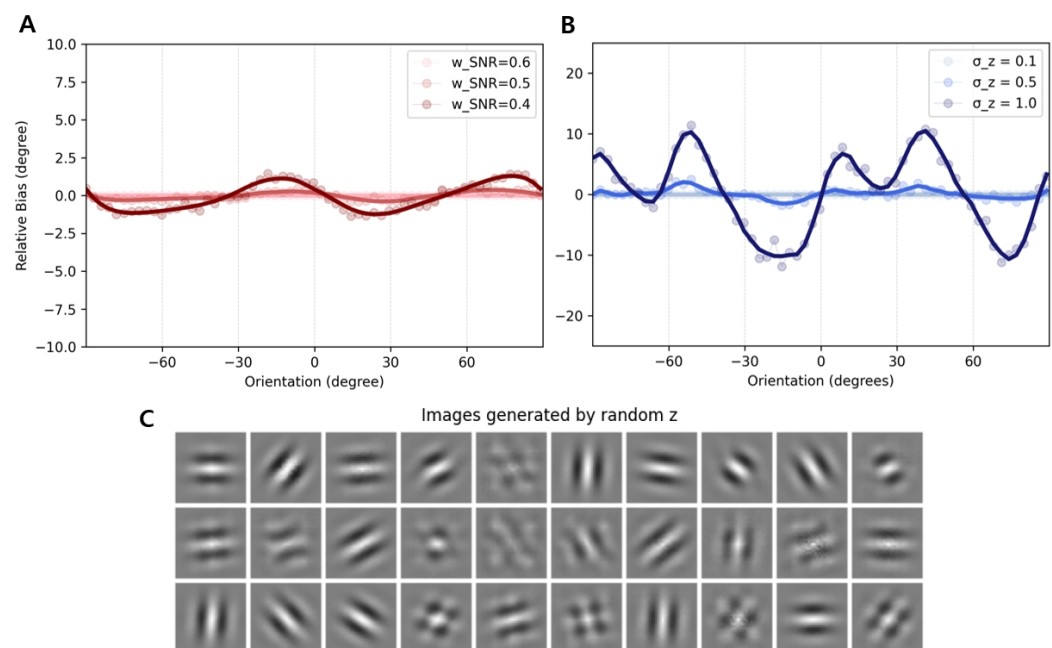

Figure A1: Ablation with reconstruction-only training. (A) Attractive bias is weakly present, but (B) repulsive bias fails to emerge, showing irregular bumps rather than systematic patterns. (C) Latent samples generate hallucinated structures that deviate from trained stimuli, indicating that adversarial training is critical for shaping a stable orientation manifold.

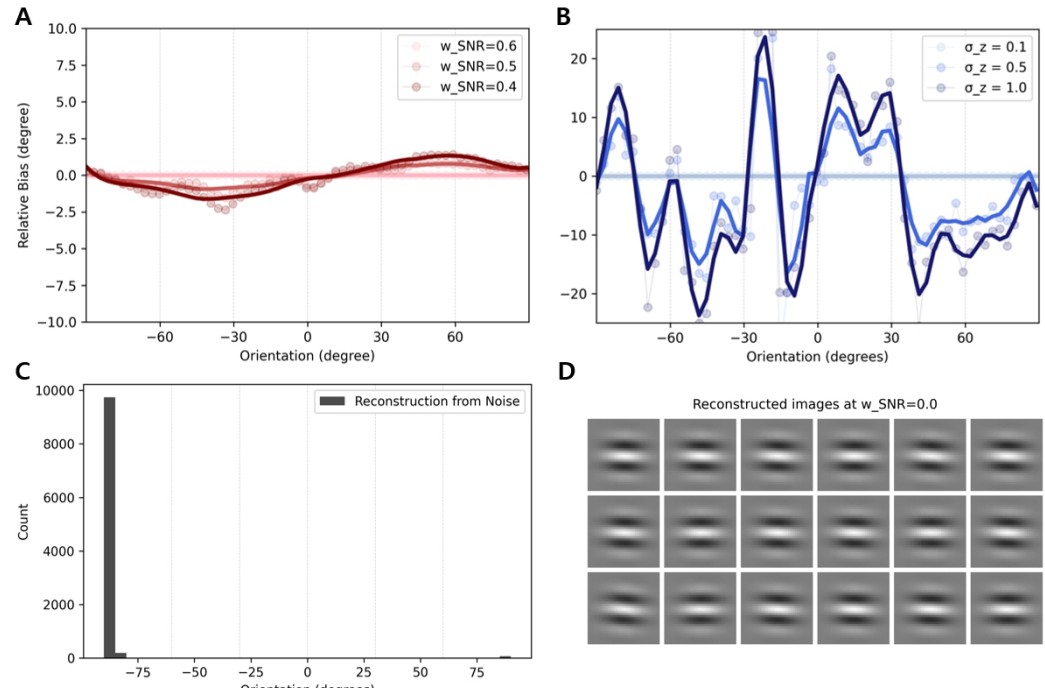

Figure A2: Ablation with adversarial-only training. (A) Attractive bias profile under stimulus noise shows residual prior-driven shifts, but (B) repulsive bias fails to emerge systematically. (C) Reconstructed orientations from maximally noisy inputs collapse almost entirely to the horizontal axis, indicating mode collapse. (D) Example Gabor reconstructions from maximally noisy inputs likewise collapse to horizontal structure, further confirming mode collapse.

