# OpenReview forum: "Attractive and Repulsive Perceptual Biases Naturally Emerge in Generative Adversarial Inference"
_ICLR.cc/2026/Conference — ICLR 2026 Conference Withdrawn Submission_

### Official Review · Reviewer_9bi7 · 2025-10-26

**Soundness:** 2
**Presentation:** 3
**Contribution:** 2
**Rating:** 6
**Confidence:** 4

**Summary:**

The paper proposes a Generative Adversarial Inference (GAI) framework, which learns perceptual-estimation biases end to end through reconstruction of noisy stimuli combined with adversarial training without explicitly specifying a likelihood or a prior. The paper then investigates the potential cause of this effect (prior and efficient coding) and conducts an ablation study to demonstrate that both losses are needed to learn the perceptual-estimation biases effectively.

**Strengths:**

This paper presents an interesting study on how the GAI can learn the perceptual-estimation biases end-to-end without any prior. It made a simple adaptation from BiGAN by adding a reconstruction noise to form GAI. Though the method is only a simple adaptation, the resulting ablation study showed that the reconstruction loss actually plays a key role in learning the biases, which is quite interesting and is what I really like about this work.

**Weaknesses:**

1. The paper mainly focuses on Gabor images and doesn’t show any evidence that this could be extended to other more naturalistic domains such as natural images, despite using a GAN-based framework which should be capable of these tasks.

2. disagree with the authors that, quote,  “*However, these models do not account for an essential factor of perceptual processing—uncertainty in inputs and representations—and thus no existing framework unifies efficient representation with uncertainty-sensitive inference in a single learned system.*” For example, VAEs model the latent space using a distribution which naturally includes uncertainty in representations, As well as current diffusion models. Recent work has also showed similar approaches unifying efficient representation with uncertainty-sensitive inference in a single learned system (Malerba et al., 2024).

3. The causality of the main results needs to be tested further. The paper shows that under a particular training distribution, the model’s learned “prior” exhibits a certain shape; however, it remains unclear whether manipulating the training statistics would causally reshape the learned prior and corresponding biases.

4. The authors claim that this model provides a computational instantiation of the “generative-adversarial brain” hypothesis. However, if other  models reproduces the same biases (like latent regularized VAEs or diffusions), the support would be severely weakened.

5. The paper claims to reproduce human data (Fig. 2 with redraw from Wei & Stocker, 2015), but only qualitatively matches it. It lacks quantitative metrics (such as bias amplitude fitting error), which may exaggerate the similarity.

Minor issues:

1. Line 371, Appendix Fig A1 actually leads to Figure 1 in the main text.

2. WGAN-GP is not cited though it is explicitly mentioned as a referenced method.

**References:**

Malerba, S. B., Micheli, A., Woodford, M., & da Silveira, R. A. (2024). Jointly efficient encoding and decoding in neural populations. PLOS Computational Biology, 20(7), e1012240.

**Questions:**

1. Can the authors demonstrate results on natural images or more complex tasks to demonstrate generality?

2. If the training distribution is re-weighted (e.g., oversampling initially rare orientations), does the learned “prior” and the direction/magnitude of bias change accordingly? What happens under a uniform training distribution, do the biases diminish or vanish?

3. Beyond toggling losses, can the authors provide sweeps over reconstruction/adversarial weightings, alternative discriminators, and other baselines (e.g., VAE/diffusion with matched capacity and data)?

4. Given that VAEs/diffusion already model uncertainty, and some recent work has already used similar approaches unifying efficient representation with uncertainty-sensitive inference, could the authors precisely delimit what is missing there and what GAI newly provides? Does these models reproduce the same biases?

5. Can the authors provide more quantitive comparisons on the biases reproduced in GAI and in humans?

---

### Official Review · Reviewer_bM6V · 2025-10-31

**Soundness:** 2
**Presentation:** 3
**Contribution:** 2
**Rating:** 4
**Confidence:** 5

**Summary:**

The paper seeks to answer the question: how do we learn efficient representations that mirror human perceptual biases? It introduces a learned encoder–generator–discriminator architecture (GAI) that jointly optimizes reconstruction and adversarial objectives. When trained on noisy Gabor patches (drawn from a hand-crafted bimodal orientation prior), the model reproduces attractive (bias toward the prior) and repulsive (bias away from it) perceptual effects similar to those observed in human orientation judgments. This is what the authors interpret as evidence of efficient coding–like representations and Bayesian-like behavior. They say that these features can emerge spontaneously from generative learning without explicitly imposing prior structure.

While the findings are conceptually interesting, the scope of the paper is quite narrow. I am also concerned that the observed bias reversals may stem from methodological artifacts rather than genuinely Bayesian inference (more below). The model indeed behaves “Bayes-like,” in that denoising pulls estimates toward high-density regions, but it is not clearly a model of efficient coding combined with Bayesian inference, as suggested throughout the text. Demonstrating that the latent representation itself carries Bayesian structure (rather than the decoded reconstructions) would make the claim much stronger. As it stands, the work offers an intriguing proof of concept but remains limited in novelty and generalizability.

**Strengths:**

1. The paper addresses an important question in perceptual modeling. The setup is conceptually clean and connects well to long-standing debates around Bayesian and efficient-coding accounts of perception.

2. The experimental setup is clear and minimal, making it easy to interpret the results. The paper describes the pipeline, objectives, and data generation process transparently, and the figures are well thought out.

3. The paper uses useful metrics/probes (such as the Fisher information peak)

4. The model architecture, losses, and training dynamics are described in sufficient detail for replication. The choice to use simple, controllable stimuli (Gabor patches) makes the phenomena interpretable and easy to relate to established psychophysical paradigms.

5. The paper provides a compelling conceptual framing that links perceptual biases, inference under uncertainty, and adversarial learning, which could inspire broader discussion in the field about how perceptual constraints emerge in learned systems.

**Weaknesses:**

1. My biggest concern is that study is limited to 32×32 Gabor patches generated from a manually defined bimodal prior. This setup is far removed from naturalistic perception and restricts the generality of the findings. As it stands, the work feels more like a proof-of-concept demonstration than a full investigation of how perceptual biases emerge in learned systems

2. The model’s bias reversals may arise from properties of the decoding procedure rather than genuinely Bayesian inference. “External noise” is implemented as additive pixel noise (not ensemble variability), and orientation is decoded through template matching on reconstructed images. Both choices that can naturally produce central-tendency or repulsion effects. These design choices make it difficult to isolate whether the biases stem from the learned representation or the readout mechanism.

3. The paper frequently refers to the model as capturing “efficient coding combined with Bayesian inference,” but there is little evidence that the internal representation itself is Bayesian. The results primarily reflect denoising behavior rather than the representation explicitly encoding uncertainty or prior beliefs.

4. The paper provides an intriguing computational analogy to human bias patterns but stops short of demonstrating new principles of perceptual inference or learning. It shows that a simple generative system can mimic bias trends, but not that it explains why or how those biases may emerge in biological systems.

**Questions:**

1. How general are the findings given the resolution and bimodal prior? Would the same effects emerge with more naturalistic or higher-dimensional stimuli?

2. Could the observed bias reversals arise from the decoding procedure rather than the learned representation? Specifically, how do additive pixel noise and template-matching readouts influence the direction of bias?

3. What evidence supports the claim that the internal representation is Bayesian? Do the latent variables explicitly encode uncertainty or prior structure, or are the results mainly a consequence of denoising toward high-density regions?

4. Beyond reproducing qualitative bias trends, what new principle of perceptual inference or learning does the model reveal?

---

### Official Review · Reviewer_fcJm · 2025-11-01

**Soundness:** 3
**Presentation:** 3
**Contribution:** 2
**Rating:** 4
**Confidence:** 4

**Summary:**

The paper aims to build a framework that can model the phenomena of attractive and repulsive perceptual biases (attract to the prior in the case of high sensory input noise, and repulse away from the prior in the case of high latent noise) without hand-crafted prior or likelihood. The paper introduces Generative Adversarial Inference (GAI) model, which includes an encoder, a generator, and a discriminator network, and trains via a combined objective function of (1) reconstruction loss and (2) adversarial generative discrimination loss (similar to GAN). The training dataset is synthetic grayscale Gabor patches with bimodal prior with injected Gaussian noise. The paper shows that GAI can replicate the attractive and repulsive bias, and ablation studies show that the model needs both the reconstruction and adversarial components to replicate the attractive and repulsive phenomenon.

**Strengths:**

The paper has the following strength:
1. Address a well-defined problem that is how to build a model to explain the attractive and repulsive perceptual bias without any hand-crafted priors or likelihood
2. The experimental results are clear and show that GAI can be trained from raw data and can replicate the attractive and repulsive phenomenon (Fig. 2, 3), and the model can reconstruct the prior under high sensory noise. The paper also includes ablation study to show the importance of having both the reconstruction and adversarial term in the loss function
3. The paper is well-written and well-organized, with clear outline to help the readers navigate, and clear background to help the readers understand the Bayesian observer framework and how the attractive and repulsive phenomenon can be explained under this framework, and the challenge of building a model in which these priors can emerge from the data, instead of hand-crafted.

**Weaknesses:**

1. While the problem is well-defined and the experimental results are clear and easy to understand, the scope of the experimental results may be quite narrow, since the training dataset is very simple (synthetic data of grayscale Gabor patch with bimodal priors). Since Generative Adversarial training usually has problem with scalability and mode collapsing, I'm curious whether the GAI framework can still work in the settings with larger dataset and multi-modal distribution. While I understand that it might be difficult to analyze perceptual biases in real dataset (e.g., CIFAR-10, ImageNet, etc.), a next step can be to add more factors into the synthetic dataset besides orientations, for examples, shape (the covariance), and also increase the number of modes to test whether GAI can be applied to any settings beyond this toy dataset. At this stage of results, I think the paper would be a solid submission to a workshop at ICLR, and it has promising potential for the main venue if the ideas and framework is being further validated with more realistic dataset with larger scale and variety.

**Questions:**

Questions
1. . (L256) What is the SNR distribution of the training samples here?

Suggestion
1. In Fig. 3, add the bias estimation to make it clearer

---

### Official Review · Reviewer_W9Lx · 2025-11-02

**Soundness:** 3
**Presentation:** 4
**Contribution:** 3
**Rating:** 4
**Confidence:** 3

**Summary:**

This paper proposes Generative Adversarial Inference, a learning paradigm that resurfaces human perceptual biases and behaviors but without explicitly encoding them into models. In short, the paradigm consists of an encoder, a generator, and a discriminator: the three are trained with both (a) a GAN-like discrimination loss (so that the discriminator can't distinguish between generated examples and their seed vs real examples and their encoding); (b) a standard reconstruction loss between the encoder and the generator. They show that the models resulting from this simple architecture naturally exhibits similar perceptual biases to ones that have been experimentally catalogued about humans.

**Strengths:**

The paper is very well-written - I am by no means an expert in this field, but I found the exposition clear and the experiments natural. The technique is novel to my knowledge, and I think it's quite interesting that a "fully learned" approach recovers cognitive biases actually exhibited by humans without hard-coding these biases in.

**Weaknesses:**

Most of my concerns are about the experimental setup of the paper:

1. The paper is restricted to very structured 2D inputs. The authors acknowledge this in the limitations section, but I didn't fully understand why extending this to color images (even CIFAR-10-sized images) would be difficult.
2. Most of the comparisons to human studies seem to be rather qualitative, it would be nice to have more quantitative results matching the two biases.
3. As far as I'm aware, the Gabor patch dataset used here is essentially one-dimensional (i.e., the encoder just needs to learn the orientation), which might limit the general applicability of the results.
4. The paper makes some appeals to biological plausibility, but they weren't really clear enough for me to understand why the proposed framework is more biologically plausible than anything else/where the basis of generator, encoder, discriminator are (or whether the authors even intended to make a claim about biological plausibility)

**Questions:**

- Did the authors perform any hyperparameter robustness or sensitivity studies? How were choices such as the size of the latent dimension chosen?

---

### Note · Authors · 2025-11-28

I have read and agree with the venue's withdrawal policy on behalf of myself and my co-authors.